# GraphQNTK: Quantum Neural Tangent Kernel for Graph Data

**Yehui Tang, Junchi Yan**[*]
Department of Computer Science and Engineering
MoE Key Lab of Artificial Intelligence
Shanghai Jiao Tong University
{yehuitang, yanjunchi}@sjtu.edu.cn

## Abstract

Graph Neural Networks (GNNs) and Graph Kernels (GKs) are two fundamental tools used to analyze graph-structured data. Efforts have been recently made in developing a composite graph learning architecture combining the expressive power of GNNs and the transparent trainability of GKs. However, learning efficiency on these models should be carefully considered as the huge computation overhead. Besides, their convolutional methods are often straightforward and introduce severe loss of graph structure information. In this paper, we design a novel quantum graph learning model to characterize the structural information while using quantum parallelism to improve computing efficiency. Specifically, a quantum algorithm is proposed to approximately estimate the neural tangent kernel of the underlying graph neural network where a multi-head quantum attention mechanism is introduced to properly incorporate semantic similarity information of nodes into the model. We empirically show that our method achieves competitive performance on several graph classification benchmarks, and theoretical analysis is provided to demonstrate the superiority of our quantum algorithm. Source code is available at https://github.com/abel1231/graphQNTK.

## 1 Introduction

Fusing quantum computing and classic machine learning has become a promising subject of research. Quantum-based algorithms have been proposed in recent years, from naive quantum non-parametric machine learning [52, 36, 43, 32] to classic-quantum hybrid deep leaning [7, 10, 46, 37, 14]. Despite that quantum machine learning (QML) has shown its potential in many machine learning tasks, quantum computing for graph learning is still in its early stage [61]. Inspired by the two popular classes of methods for learning on graph data, i.e., Graph Neural Networks (GNNs) [20, 39, 21, 67] and Graph Kernels (GKs) [23], several works attempt to build quantum graph learning architecture that captures the structural information of graph data, such as Quantum Graph Neural Networks (QGNNs) [64, 7, 11, 16, 1] and Quantum Graph Kernel Methods (QGKs) [56, 3, 25, 4]. A brief review about quantum graph learning is illustrated in Fig. 1.

Some quantum subroutines for attribute encoding [5, 70] and structural encoding [64, 46] have been developed to dissolve the characteristics of the graph into the quantum model. However, most present quantum graph learning models are hybrid such that the expressive capability depends more on the complexity of the classic modules [70]. It is difficult to characterize the structure information and attribute information of the graph by the quantum components without the participation of classic modules. Even worse, the frequent interactions between classical systems and quantum environments generally incur additional overhead [55]. It is unclear whether the introduced quantum module

---

[*]Junchi Yan is the correspondence author who is also with Shanghai AI Laboratory.

36th Conference on Neural Information Processing Systems (NeurIPS 2022).

can improve the performance of the model as well as the training efficiency. Besides, most of existing proposals for quantum machine learning for graphs lack a clear demonstration of a quantum superiority for tasks on classical datasets.

Using quantum computing power to boost the trainability and expressive behaviour of classic machine learning models provides one of the most promising direction for quantum machine learning. It is demonstrated that the power of quantum computing could be used to find atypical but useful patterns that classical systems are not considered to be able to generate effectively [14, 24, 28], and accelerate the training process of existing classic models [36, 43, 71]. Several quantum algorithms [52, 45, 44] based on the HHL algorithm [22] show the exponential speedup compared with their classical counterpart, with a assumption that a quantum random access memory (QRAM) [35] is accessible. Recent literature employ quantum algorithms to efficiently train deep neural networks [71], reconstruct unsupervised clustering [34] and supervised kernel classifier [43]. It is hopeful that quantum computing could provide a new learning paradigm. In addition, simulations and physical experiments have proved the potential of using quantum algorithms to encode and process regular classical data such as text and image [60, 6].

Beyond vanilla GNNs and GKs, composite graph learning studies have emerged that combine the advantages of both areas [50, 10, 18]. In this literature, Graph Neural Tangent Kernel (GNTK) [17] based on neural tangent kernel (NTK) [31] shines its lights on elegantly fusing GNNs and GKs, leading to new prospectives of training and analysing infinite-width GNNs. However, the computation overheads is extremely large due to either the dense gram matrix [17], or the large number of substructures to be compared after graph decomposition [10].

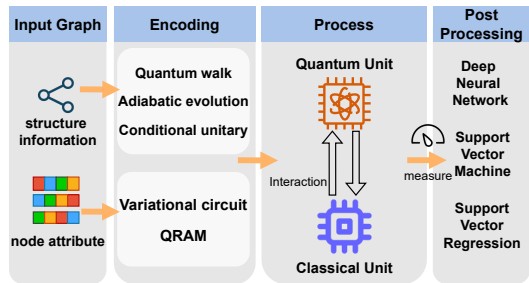

Figure 1: Overview of quantum graph learning.

Prospectively, the barrier that conventional model is difficult to train and scale up is expected to be circumvented with the help of the uniqueness of quantum computing. Early research involves altering the amplitude of quantum basis states to accomplish a quantum logic operations [8], which is profitable from the huge quantum Hilbert space to encode the normalized data. Recently, simultaneous transformation of basic states in quantum superposition using quantum parallelism is regarded as a remarkable manifestation of quantum superiority, which is successfully implemented in classic machine learning to reduce the computational overheads [36, 37, 71]. These strategies could be helpful in the regime of training graph models with either the non-convex nature of the training procedure, or the poor scalability w.r.t. training size.

In this paper, we focus on quantum machine learning of graph-structured data with attributed nodes and binary edges. Inspired by recent quantum neural network methods [37, 71] that efficiently reconstruct the dynamics of classic neural networks using quantum computing techniques, a new quantum graph learning model is proposed which is analogue to train an infinite-width GNN with attention mechanism, where the number of heads goes to infinity. Our contributions are:

- **Attention-enhanced GNTK.** Infinite-width GNN is a well established term in the GNN literature [17, 29], and GNTK [17] is a powerful tool to analyze the GNN. However, the traditional feature aggregation of the vanilla GNTK is straightforward, limiting itself within joint neighborhoods. In this paper, a multi-head attention mechanism is introduced to properly incorporate semantic similarity information of disconnected nodes but with similar features to improve model expressiveness.

- **Kernel methods for evaluating the dynamics of wide and deep GNNs.** It is generally hard to train a deep GNN with attention, especially when the width of GNN, the width of attention layer, or the number of heads goes to large. We properly incorporate infinite-width multi-head attention into GNTK by using NTK theory. We use kernel methods to capture the dynamics of infinite-width GNN with infinite-width attention, thus avoiding the huge overhead of training a wide and deep GNN.

- **Speedup introduced by quantum computing.** Although GNTK is a useful method to train an infinite-width GNN, its cost still grows quadratically with respect to the volume of data, which is intractable for large datasets. We re-design the Attention-enhanced GNTK by splitting it into small components and reuniting them using the quantum linear algebra subroutines. The produced

quantum graph kernel – GraphQNTK, theoretically reduces the computational complexity from $O(N^2)$ to $O(N)$ benefited from the quantum parallelism.

## 2 Methodology

### 2.1 Preliminaries

We first briefly review the most common setting for GNNs and the corresponding NTK, and by the way the notation is given. A graph $G = (V, E)$ is denoted by a collection of nodes $V$ and edges $E$. Each node has a $d$-dimensional feature vector $\mathbf{h}_v \in \mathbb{R}^d$, $v \in V$, and $\mathbf{H} \in \mathbb{R}^{n \times d}$ is the feature matrix stacking all nodes features. For graph classification, we consider the dataset with a set of graphs $\{G_1, \ldots, G_N\} \subseteq \mathcal{G}$ and their labels $\{y_1, \ldots, y_N\} \subseteq \mathcal{Y}$. Our goal is to learn to predict labels of unseen graphs.

**The formulation of GNN**. The differences of GNNs mainly depend on the different settings of message propagation process. Here we consider a simple message passing framework [20] and the propagation of the $l$-th ($l \in [L]$) layer is given as:

$$\hat{\mathbf{h}}_u^l = \sum_{v \in \mathcal{N}(u) \cup \{u\}} \mathbf{h}_v^{(l-1)}, \tag{1}$$

$$\mathbf{h}_u^l = \sqrt{\frac{c_\sigma}{d_R^l}} \sigma \left( \mathbf{W}_R^l \sqrt{\frac{c_\sigma}{d_{R-1}^l}} \sigma \left( \mathbf{W}_{R-1}^l \cdots \sqrt{\frac{c_\sigma}{d_1^l}} \cdot \sigma \left( \mathbf{W}_1^l \hat{\mathbf{h}}_u^l \right) \right) \right), \tag{2}$$

where $\mathcal{N}(u)$ denotes the neighbors of $u$, $c_\sigma$ is the scaling factor, $d_r^l$ is the output dimension of the $l$-th layer and the $r$-th fully-connected layers, $\sigma$ is an element-wise activated function, and $\mathbf{W}_R^l$ is learnable weights performing on the input for $R$ times of the $l$-th layer (equivalent to $R$ fully-connected layers without the bias term).

For graph classification, the output is a permutation invariance function acting on the collection of all node features in the last layer. The popular `sum_pooling` function is adopted: $\mathbf{h}_G = \sum_{u \in V} \mathbf{h}_u^L$..

**NTK of the infinite-width GNN.** Consider a training set $\{(\mathbf{x}_i, y_i)\}_{i=1}^N \subset \mathbb{R}^d \times \mathbb{R}$. When an over-parameterized fully connected network $f(\theta, \mathbf{x}) : \mathbb{R}^d \to \mathbb{R}$ whose width is allowed to go to infinity and parameters $\theta$ are randomly initialized and trained with gradient descent, the dynamics of the network is equivalent to the kernel regression [31]. This is the so called neural tangent kernel (NTK):

$$\mathbf{H}(t)_{ij} = \left\langle \frac{\partial f(\theta(t), x_i)}{\partial \theta}, \frac{\partial f(\theta(t), x_j)}{\partial \theta} \right\rangle, \tag{3}$$

which remains constant during training, i.e., $\mathbf{H}(t) = \mathbf{H}(0)$. And we replace $\mathbf{H}(t)$ with $\mathbf{H}$ for convenience. The final prediction for a test datapoint $\mathbf{x}_*$ is

$$f(\mathbf{x}_*) = \mathbf{k}_* \mathbf{H}^{-1} \mathbf{y}, \tag{4}$$

where $\mathbf{y}_i = y_i$ and $\mathbf{k}_* \in \mathbb{R}^N$ is the vector whose $i$-th element denotes the NTK value between $\mathbf{x}_i$ and $\mathbf{x}_*$.

It is discovered that convolutional neural networks (CNNs) with infinite-width channels and infinite number of filters also have the same behaviour [2]. Inspired by this, Du et al. [17] adopts the designing strategy of NTK and leverages a GNN architecture to design new graph kernels, which is called GNTK. The dynamics of training the GNTK is equivalent to train an infinitely-wide GNN initialized with random weights trained with gradient descent. Specifically, consider two input graph $G = (V, E)$ and $G' = (V', E')$ with $|V| = n$ and $|V| = n'$, the GNTK $\boldsymbol{\Theta} \in \mathbb{R}^{n \times n'}$ and the relative covariance matrix $\boldsymbol{\Sigma} \in \mathbb{R}^{n \times n'}$ in the $l$-th layer of the feature aggregation phase as described in Eq. 1 after $R$ fully-connected layers are given by

$$\left[ \boldsymbol{\Sigma}_0^l (G, G') \right]_{uu'} = \sum_{v \in \mathcal{N}(u) \cup \{u\}} \sum_{v' \in \mathcal{N}(u') \cup \{u'\}} \left[ \boldsymbol{\Sigma}_R^{l-1} (G, G') \right]_{vv'},$$

$$\left[ \boldsymbol{\Theta}_0^l (G, G') \right]_{uu'} = \sum_{v \in \mathcal{N}(u) \cup \{u\}} \sum_{v' \in \mathcal{N}(u') \cup \{u'\}} \left[ \boldsymbol{\Theta}_R^{l-1} (G, G') \right]_{vv'}, \tag{5}$$

which is an affine transformation of the input GNTK and covariance respectively where $\left[\mathbf{\Theta}_R^0(G, G')\right]_{uu'}$ and $\left[\mathbf{\Sigma}_R^0(G, G')\right]_{uu'}$ are both defined to be $\mathbf{h}_u^\top \mathbf{h}_{u'}$. We replace them with $\left[\mathbf{\Theta}^0(G, G')\right]_{uu'}$ and $\left[\mathbf{\Sigma}^0(G, G')\right]_{uu'}$ respectively without ambiguity.

The successive fully-connected layers defined in Eq. 2 are used to update the node hidden feature after aggregation. Specifically, the GNTK of the fully-connected layer is recursively associated to that of the previous layer, and the transformation is given by

$$\left[\mathbf{\Sigma}_r^l(G, G')\right]_{uu'} = \hat{\sigma}^{(r-1)}\left(\left[\mathbf{\Sigma}_{r-1}^l(G, G')\right]_{uu'}\right), r \in [R], \tag{6}$$

where $\hat{\sigma}^{(r)} : [-1, 1] \to \mathbb{R}$ denotes the the conjugate activation function corresponding to the activated function $\sigma$ with centered Gaussian processes of covariance at the $r$-th fully-connected layer, as described in [15]. And the derivation of the covariance is

$$\left[\dot{\mathbf{\Sigma}}_r^l(G, G')\right]_{uu'} = \hat{\dot{\sigma}}\left(\hat{\sigma}^{(r-1)}\left(\left[\mathbf{\Sigma}_{r-1}^l(G, G')\right]_{uu'}\right)\right), \tag{7}$$

where $\hat{\dot{\sigma}}$ denotes the derivative of $\sigma$. Given Eq. 6 and Eq. 7, the transformation of the GNTK for the feature update phase denoted by Eq. 2 is given by

$$\left[\mathbf{\Theta}_R^l(G, G')\right]_{uu'} = \sum_{r=1}^R \left[\mathbf{\Sigma}_0^l(G, G')\right]_{uu'}\left(\prod_{r'=r}^R \left[\dot{\mathbf{\Sigma}}_0^l(G, G')\right]_{uu'}\right). \tag{8}$$

Therefore, computing each element of the GNTK (or covariance) matrix is only reliant on the element at the same place of the GNTK (or covariance) matrix in the previous fully-connected layer. The final GNTK corresponding the two input graphs $G$ and $G'$ determined by the `sum_pooling` function:

$$\mathbf{\Theta}(G, G') = \sum_{u \in V, u' \in V'} \left[\mathbf{\Theta}_R^L(G, G')\right]_{uu'}. \tag{9}$$

Intuitively, calculating each element of the GNTK of fully-connected layers could be accelerating by a proper quantum kernel estimation algorithm. However, it is indirect to realize an end-to-end speedup for GNTK since calculating the element of GNTK requires an affine transformation. To circumvent this barrier, we derive a unitary quantum aggregation transformation to bridge the gap between quantum kernel methods and estimation of GNTK.

## 2.2 QNTK with Attention Mechanism

Before giving the analytical quantum reconstruction of GNTK with multi-head attention mechanism, we first elaborate on how to integrate the transformer layer into the GNN as described in Sec. 2.1. The resulting GraphQNTK can be efficiently reconstructed by quantum computing paradigm, which gives a quadratic speed-up over the classic estimation of GNTK. The mechanism to build the GNN and estimate the GNTK is shown in Fig. 2.

**GNN with multi-head attention.** The aggregation process of vanilla GCN [39] regards the contribution of each node's neighbor to the central node as equally important, which can be viewed as learning an averaged filter across the whole graph [66], leading to a great loss of structure information. Besides, the aggregation only is performed within the adjoining neighbors under the assumption that the graph is homophilous. The method may fail to learn effective graph structures for message passing [12]. To capture the global node similarity semantics of the provided graph, numerous attempts that employ transformer for graph learning have been developed [27, 51, 53, 68]. Consider the input feature matrix $\mathbf{H}_{\text{in}}^l \in \mathbb{R}^{N \times s^l}$ where $N$ denotes the number of samples and $s^l$ is the dimension of feature at layer $l$ before implementation of the transformer. The single transformer layer is to project the input $\mathbf{H}_{\text{in}}^l \in \mathbb{R}^{N \times s^l}$ by three matrices, i.e., $\mathbf{W}_Q^l \in \mathbb{R}^{s^l \times s_K^l}$, $\mathbf{W}_K^l \in \mathbb{R}^{s^l \times s_K^l}$ and $\mathbf{W}_V^l \in \mathbb{R}^{s^l \times s_V^l}$, to the corresponding representations $\mathbf{Q}^l, \mathbf{K}^l, \mathbf{V}^l$. The formulation is given as

$$\mathbf{Q}^l = \mathbf{H}_{\text{in}}^l \mathbf{W}_Q^l, \quad \mathbf{K}^l = \mathbf{H}_{\text{in}}^l \mathbf{W}_K^l, \quad \mathbf{V}^l = \mathbf{H}_{\text{in}}^l \mathbf{W}_V^l,$$

$$\widehat{\mathbf{H}}^l = \zeta\left(\mathbf{G}^l\right)\mathbf{V}^l, \quad \mathbf{G}^l = \frac{\mathbf{Q}^l \mathbf{K}^{l^\top}}{\sqrt{s_K^l}}, \tag{10}$$

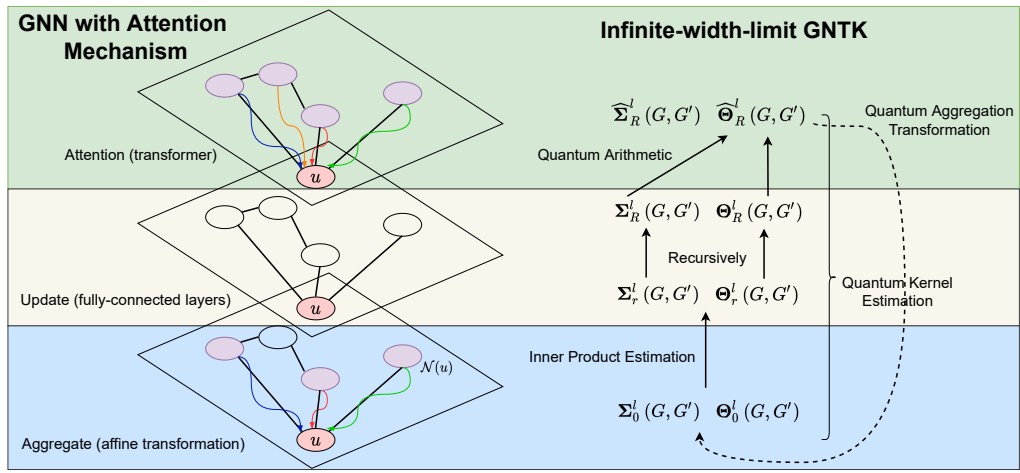

Figure 2: Framework for GNN with attention mechanism and its corresponding GNTK. The GNN comprises a message transmission process similar to the vanilla GCN but involves a transformer at the tail of the model (excluding the last layer), which characterizes the global semantic similarity between each pairs of nodes. The neighbor aggregation is kept since the two nodes connected by an edge often have stronger semantic relationship. The dynamics of the infinite-width-limit GNN is analogous to kernel methods and we reconstruct it by quantum algorithms to estimate the kernel.

where $\zeta$ denotes an element-wise activated function. The multi-head attention alternative is given by

$$\mathbf{H}_{\text{out}}^l = \left[\widehat{\mathbf{H}}_{\text{head}_1}^l, \dots, \widehat{\mathbf{H}}_{\text{head}_M}^l\right] \mathbf{W}_O^l, \tag{11}$$

where $\mathbf{W}_O^l \in \mathbb{R}^{(Ms_V^l)\times s^l}$ projects the $N \times Ms_V^l$ concatenated multi-head feature matrix back to $N \times s^l$ matrix.

Let $\mathbf{Y}$ and $\mathbf{\Theta}$ denote the neural network Gaussian Process Kernel (NNGP) [51] and NTK after the transformer layer, and let $\widetilde{\mathbf{Y}}$ and $\widetilde{\mathbf{\Theta}}$ be the input NNGP and NTK before the transformer layer. Consider two input feature vector $\mathbf{x}$ and $\mathbf{x}'$. When the output dimension of the transformer layer and the number of heads go to infinity, i.e., $s^l \to \infty, s_K^l \to \infty, s_V^l \to \infty, M \to \infty$, the output NTK is:

$$\mathbf{\Theta}(\mathbf{x}, \mathbf{x}') = 2\mathbf{Y}(\mathbf{x}, \mathbf{x}') + \zeta\left(\widetilde{\mathbf{Y}}(\mathbf{x}, \mathbf{x})\right)\widetilde{\mathbf{\Theta}}(\mathbf{x}, \mathbf{x}')\zeta\left(\widetilde{\mathbf{Y}}(\mathbf{x}', \mathbf{x}')\right)^\top,$$

$$\mathbf{Y}(\mathbf{x}, \mathbf{x}') = \zeta\left(\widetilde{\mathbf{Y}}(\mathbf{x}, \mathbf{x})\right)\widetilde{\mathbf{Y}}(\mathbf{x}, \mathbf{x}')\zeta\left(\widetilde{\mathbf{Y}}(\mathbf{x}', \mathbf{x}')\right)^\top, \tag{12}$$

where the under the restriction that 1) $\mathbf{W}_Q^l$ and $\mathbf{W}_K^l$ share the same weighs, and 2) scaling the dot products between $\mathbf{Q}^l$ and $\mathbf{K}^l$ by their dimension instead of the square root of the same quantity, i.e., $\mathbf{G}^l = \frac{\mathbf{Q}^l \mathbf{K}^{l\top}}{s_K^l}$. The detailed proof can be found in [26].

To efficiently estimate the element of the NTK defined by the transformer layer using quantum parallel, we slightly modify the Eq. 12 as

$$\mathbf{\Theta}(\mathbf{x}, \mathbf{x}') = 2\mathbf{Y}(\mathbf{x}, \mathbf{x}') + \widetilde{\mathbf{T}}_\zeta(\mathbf{x}, \mathbf{x}') \odot \widetilde{\mathbf{\Theta}}(\mathbf{x}, \mathbf{x}'),$$

$$\mathbf{Y}(\mathbf{x}, \mathbf{x}') = \widetilde{\mathbf{T}}_\zeta(\mathbf{x}, \mathbf{x}') \odot \widetilde{\mathbf{Y}}(\mathbf{x}, \mathbf{x}'), \tag{13}$$

where $\widetilde{\mathbf{T}}(\mathbf{x}, \mathbf{x}')$ is the result of matrix multiplication between the column vector of the diagonal of $\widetilde{\mathbf{Y}}(\mathbf{x}, \mathbf{x})$ and row vector of the diagonal of $\widetilde{\mathbf{Y}}(\mathbf{x}', \mathbf{x}')$, and $\widetilde{\mathbf{T}}_\zeta$ is the result of matrix multiplication between the diagonal of those two matrix after activated operation. For simplicity of use, we consider identity function as the activated operation, i.e, $\zeta = I$. It is reasonable to accept this modification since in the limit of infinite width neural network the output converges in distribution to a multivariate normal with a block diagonal covariance [51]. Notice that the difference between the definition of NNGP and the covariance of NTK is that the former denotes the expectation with respect to the output before the activated operation, while the later after the denotes the expectation with respect to the output after the activated operation [31]. Consequently, we consider that $\mathbf{Y}$ is equal to the covariance of NTK within the transformer layer as the result of the identity activated function.

**GNTK with infinite-width-limit attention.** To appropriately incorporate semantic similarity information of nodes into the model, a multi-head attention mechanism is implemented at the tail of the each GNN layer except the first and the last layer, and the calculation of the GNTK with infinite-width-limit attention is to insert an additional procedure after the fully connected layers. For the two input graphs $G$ and $G'$, the formulation derived by Eq. 13 is given as

$$
\begin{aligned}
\left[\widehat{\mathbf{\Theta}}_R^l(G,G')\right]_{uu'} &= 2\left[\widehat{\mathbf{\Sigma}}_R^l(G,G')\right]_{uu'} + \left[\mathbf{T}^l(G,G')\right]_{uu'}\left[\mathbf{\Theta}_R^l(G,G')\right]_{uu'}, \\
\left[\widehat{\mathbf{\Sigma}}_R^l(G,G')\right]_{uu'} &= \left[\mathbf{T}^l(G,G')\right]_{uu'}\left[\mathbf{\Sigma}_R^l(G,G')\right]_{uu'},
\end{aligned}
\tag{14}
$$

where $\mathbf{T}^l(G,G')$ is the result of matrix multiplication between the column vector of the diagonal of $\mathbf{\Sigma}_R^l(G,G)$ and row vector of the diagonal of $\mathbf{\Sigma}_R^l(G',G')$. The affine transformation of the input GNTK corresponding to the aggregation phase as described in Eq. 5 is changed to (similar to $\mathbf{\Sigma}_0^l$):

$$
\left[\mathbf{\Theta}_0^l(G,G')\right]_{uu'} = \sum_{v\in\mathcal{N}(u)\cup\{u\}}\sum_{v'\in\mathcal{N}(u')\cup\{u'\}}\left[\widehat{\mathbf{\Theta}}_R^{l-1}(G,G')\right]_{vv'},
\tag{15}
$$

### 2.3 The Proposed GraphQNTK

We first show that estimating the single-layer GraphQNTK and its covariance with infinite-width-limit attention mechanism can be efficiently reconstructed in the regime of quantum computing, and generalize to the multi-layer model. The following statements only consider two input graphs $G = (V,E)$ and $G' = (V',E')$ with $|V| = n$ and $|V'| = n'$, and the corresponding feature matrix $\mathbf{H} = [\mathbf{h}_1^\top,\cdots,\mathbf{h}_u^\top,\cdots,\mathbf{h}_n^\top] \in \mathbb{R}^{n\times d}$ and $\mathbf{H}' = [\mathbf{h}_1^\top,\cdots,\mathbf{h}_{u'}^\top,\cdots,\mathbf{h}_{n'}^\top] \in \mathbb{R}^{n'\times d}$. The approximate estimation of GNTK is denoted as $\bar{\mathbf{\Theta}} \in \mathbb{R}^{n\times n'}$ and its element is $\bar{\mathbf{\Theta}}_{uu'}$. The corresponding covariance is $\bar{\mathbf{\Sigma}} \in \mathbb{R}^{n\times n'}$ and $\bar{\mathbf{\Sigma}}_{uu'}$. We use $\bar{\mathbf{\Theta}}_{GG'} \in \mathbb{R}$ to represent GraphQNTK after readout. We omit the subscript $R$ for clarity. The same setting can be easily generalized to the arbitrary pair of graphs $G,G' \in \mathcal{G}$ by introducing auxiliary index registers. First, we introduce the quantum data structure accessible to the classical data, as commonly used by QML algorithms [52, 36, 34, 71].

**Feature encoding.** Using the storage structure as stated in our proposed Theorem 1 in Appendix, the feature matrix can be prepared into the QRAM at the initialization of the algorithm. The data encoding only occurs a single time and readout operation only takes logarithmic complexity time with respect to the number of samples $n$ and dimension of feature $d$. The quantum representations corresponding to the encoded feature vector and feature matrix are as follows

$$
\begin{aligned}
|u\rangle|0\rangle \to |u\rangle|\mathbf{h}_u\rangle, &\quad |0\rangle \to \frac{1}{\|\mathbf{H}\|_F}\sum_u\|\mathbf{h}_u\||u\rangle, \\
|u'\rangle|0\rangle \to |u'\rangle|\mathbf{h}_{u'}\rangle, &\quad |0\rangle \to \frac{1}{\|\mathbf{H}'\|_F}\sum_{u'}\|\mathbf{h}_{u'}\||u'\rangle.
\end{aligned}
\tag{16}
$$

**Estimation of the initialized NTK.** The empirical uncentered covariance of inputs $\left[\mathbf{\Sigma}^0(G,G')\right]_{uu'}$ and the initialized GNTK $\left[\mathbf{\Theta}^0(G,G')\right]_{uu'}$ is the inner product between $\mathbf{h}_u$ and $\mathbf{h}_{u'}$. Following a similar approach to [37], the inner product between two vectors with respect to their quantum representations can be estimate efficiently by introducing an auxiliary register. Specifically, estimation of the inner product $\mathbf{h}_u^\top\mathbf{h}_{u'}$ can be performed by constructing the state $\frac{1}{\sqrt{2}}(|u\rangle|u'\rangle|0\rangle||\mathbf{h}_u\rangle\rangle + |u\rangle|u'\rangle|1\rangle||\mathbf{h}_{u'}\rangle\rangle)$. Applying a Hadamard gate on the third register gives the state $|u\rangle|u'\rangle\left(\sqrt{P_{uu'}}|0,g_{uu'}\rangle + \sqrt{1-P_{uu'}}|1,g'_{uu'}\rangle\right)$, where $P_{uu'} = \frac{1+\mathbf{h}_u^\top\mathbf{h}_{u'}}{2}$ is the estimation of the inner product. This procedure takes $O(\log d)$ time and we denote this quantum operation by $\mathcal{D}^0$, and we add a subscript to denote the corresponding conditioned operator, i.e, $\mathcal{D}_{uu'}^0$ represents $\mathcal{D}^0$ is conditioned acting on the basis state coupled with state $|0\rangle \to |u\rangle|u'\rangle$. We can perform the $\mathcal{D}_{uu'}^0$ in superposition such that the state $\frac{1}{\sqrt{nn'}}\sum_{u\in V}\sum_{u'\in V'}|u\rangle|u'\rangle\left(\sqrt{P_{uu'}}|0,g_{uu'}\rangle + \sqrt{1-P_{uu'}}|1,g'_{uu'}\rangle\right)$ can be generated in time $O(\log(nd))$.

**Quantum aggregation transformation.** Recall that an affine transformation (refer to Eq. 5 and Eq. 15) acting on the GNTK and its covariance is relative to the neighborhood aggregation defined by Eq. 1. Therefore, it is indirect to realize an end-to-end speedup similar to the estimation of the inner

product since the transformation of each element of NTK and the covariance is not independent. To circumvent this barrier, we derive a unitary quantum aggregation transformation to approximately reconstruct the affine transformation. Consider the quantum operation $\mathcal{D}^0_{uu'} : |u\rangle|u'\rangle|0\rangle|0\rangle \to |u\rangle|u'\rangle \left( \sqrt{P_{uu'}}|0, g_{uu'}\rangle + \sqrt{1 - P_{uu'}}|1, g'_{uu'}\rangle \right)$ that is employed to estimate the inner product of two feature vectors. Define a unitary operator which is used to perform aggregation transformation

$$\mathcal{U} = \sum_{v \in \mathcal{N}(u)\cup\{u\}} \sum_{v' \in \mathcal{N}(u')\cup\{u'\}} |v\rangle\,|v'\rangle\,\langle v|\,\langle v'| \otimes \mathcal{D}^0_{vv'}, \tag{17}$$

which can be generated by introducing conditional quantum evolution [22]. The operation $\otimes$ denotes the tensor product. We apply the $\mathcal{U}$ with Hadamard gates to the given initial state, which is given as

$$H^{\otimes}\mathcal{U}H^{\otimes}|0\rangle^{\otimes}|0\rangle|0\rangle \to H^{\otimes}\mathcal{U}\sum_{v,v'}|v, v'\rangle\,|0\rangle|0\rangle$$

$$\to H^{\otimes}\sum_{v \in \mathcal{N}(u)\cup\{u\}}\sum_{v' \in \mathcal{N}(u')\cup\{u'\}}|v, v'\rangle\left(\sqrt{P_{vv'}}|0, g_{vv'}\rangle + \sqrt{1 - P_{vv'}}|1, g'_{vv'}\rangle\right) \tag{18}$$

$$\to \sum_{v \in \mathcal{N}(u)\cup\{u\}}\sum_{v' \in \mathcal{N}(u')\cup\{u'\}}\sqrt{P_{vv'}}|0\rangle^{\otimes} + \sqrt{\cdot}\,|\text{other}\rangle + \cdots$$

where $\sqrt{\cdot}\,|\text{other}\rangle$ represents other computational basis states except for $|0\rangle^{\otimes}$ with amplitude $\sqrt{\cdot}$, and the detailed mathematical expression and the scalar for state normalization are omitted since the result of the affine transformation has been embedded into the amplitude of $|0\rangle^{\otimes}$. The $(\cdot)^{\otimes}$ denotes that there could be multiple unitary operations acting on multiple registers, depending on the number of qubits required to encode the classic data. Similar to the inner product estimation, the quantum aggregation transformation can be performed in superposition and the resulting superposition is

$$\frac{1}{\sqrt{nn'}}\sum_{u \in V}\sum_{u' \in V'}|u\rangle|u'\rangle\left(\sqrt{A_{uu'}}|0, y_{uu'}\rangle + \sqrt{1 - A_{uu'}}|1, y'_{uu'}\rangle\right),$$

$$\sqrt{A_{uu'}} = \frac{\sum_{v \in \mathcal{N}(u)\cup\{u\}}\sum_{v' \in \mathcal{N}(u')\cup\{u'\}}\sqrt{P_{vv'}}}{|v| \times |v'|}. \tag{19}$$

The amplitude $\sqrt{A_{uu'}}$ can be encoded into an ancillary register by using Amplitude Estimation (Theorem 3) and Median Evaluation (Theorem 4). The obtained quantum state $\frac{1}{\sqrt{nn'}}\sum_{u \in V}\sum_{u' \in V'}|u\rangle|u'\rangle|\bar{A}_{uu'}\rangle|y_{uu'}\rangle$ whose third register carries the approximate result after aggregation transformation as described in Eq. 5 and Eq. 15, where $|A_{uu'} - \bar{A}_{uu'}| \leq \epsilon$ and $|y_{uu'}\rangle$ is a garbage state. The runtime is $O(\log(nd)log(1/\Delta)/\epsilon)$ and $\Delta$ is the proximity defined by the Median Evaluation. Note that $\bar{A}_{uu'}$ is actually the polynomial combination of the element-wise square root of the NTK from the previous layer, thus it is an approximate aggregation transformation. In the experiment, we empirically show that this approximation has a restrictive effect on the performance.

**Quantum kernel estimation**. For fully-connected neural network, the calculation of each element of the NTK and its covariance is only reliant on the element at the same position of the covariance matrix in the previous fully-connected layer. Besides, the affine transformation of the GNTK and its covariance can be efficiently approximated by quantum aggregation transformation and the result has been embedded into the basis states of a superposition. In general, there exits a unitary $V : \sum_x |x, 0\rangle \to \sum_x |x, f(x)\rangle$ for any classical function $f$ with the same time complexity to evaluate each element of the NTK and each element of the covariance [48, 71]. Specifically, an oracle which operates as the same as classical function defined by Eq. 8 is implemented on the third register of $\frac{1}{\sqrt{nn'}}\sum_{u \in V}\sum_{u' \in V'}|u\rangle|u'\rangle|\bar{A}_{uu'}\rangle|y_{uu'}\rangle$. The resulting NTK is $\frac{1}{\sqrt{nn'}}\sum_{u \in V}\sum_{u' \in V'}|u\rangle|u'\rangle|\bar{\Theta}_{uu'}\rangle|y_{uu'}\rangle$, where $\bar{\Theta}_{uu'}$ is the approximate estimation of its classical counterpart after $R$ fully-connected layers. The oracle is expected to be with the same complexity of its classical counterpart, which is associative to the number of fully-connected layers and is independent on the number of training samples $n$. For estimation of the GNTK after a transformer layer (Eq. 14), the covariance $\Sigma^l_R(G, G)$ for any $G \in \mathcal{G}$ requires to be estimated in advance. It means that the state $\frac{1}{n}\sum_{u \in V}\sum_{u' \in V}|u\rangle|u'\rangle|\bar{\Sigma}_{uu'}\rangle|y_{uu'}\rangle$ must be estimated for any $G(V, E) \in \mathcal{G}$ before input the different graphs, and we only consider the element when $u = u'$. By taking the partial trace on the second register, we obtain the state $\frac{1}{\sqrt{n}}\sum_{u \in V}|u\rangle|\bar{\Sigma}_{uu}\rangle|y_{uu}\rangle$ for graph $G$ and

$\frac{1}{\sqrt{n'}}\sum_{u'\in V'}|u'\rangle|\bar{\mathbf{\Sigma}}_{u'u'}\rangle|y_{u'u'}\rangle$ for graph $G'$. Thus, estimation of the GNTK and its covariance corresponding to the multiplication part in Eq. 14 is given as

$$\frac{1}{\sqrt{nn'}}\sum_{u\in V}\sum_{u'\in V'}|u\rangle|u'\rangle|\bar{\mathbf{\Theta}}_{uu'}\rangle|y_{uu'}\rangle \rightarrow \frac{1}{\sqrt{nn'}}\sum_{u\in V}\sum_{u'\in V'}|u\rangle|u'\rangle|\bar{\mathbf{\Theta}}_{uu'}\times\bar{\mathbf{\Sigma}}_{uu}\times\bar{\mathbf{\Sigma}}_{u'u'}\rangle|y_{uu'}\rangle,$$

$$\frac{1}{\sqrt{nn'}}\sum_{u\in V}\sum_{u'\in V'}|u\rangle|u'\rangle|\bar{\mathbf{\Sigma}}_{uu'}\rangle|y_{uu'}\rangle \rightarrow \frac{1}{\sqrt{nn'}}\sum_{u\in V}\sum_{u'\in V'}|u\rangle|u'\rangle|\bar{\mathbf{\Sigma}}_{uu'}\times\bar{\mathbf{\Sigma}}_{uu}\times\bar{\mathbf{\Sigma}}_{u'u'}\rangle|y_{uu'}\rangle. \tag{20}$$

This is performed by using the conditional quantum adder and the multiplier conditioned on the index register, i.e. $|u\rangle$ and $|u'\rangle$, which are designed by [63, 54, 41]. The final GNTK after the transformer layer can be directly generated by additional quantum arithmetic operations that perform an element-wise addition between the covariance to the GNTK.

**Estimation the GNTK for multiple layers**. The quantum aggregation transformation requires that the approximate NTK and its covariance are embedded into the amplitudes of a superposition. However, after the quantum kernel estimation, these matrix are embedded into the quantum basis states of a superposition. To extract them back to the amplitudes, we apply Conditional Rotation [37] on the register containing the approximate GNTK (and the covariance), which is given by

$$\frac{1}{\sqrt{nn'}}\sum_{u\in V}\sum_{u'\in V'}|u\rangle|u'\rangle|\bar{\mathbf{\Theta}}_{uu'}\rangle \rightarrow \frac{1}{\sqrt{nn'}}\sum_{u\in V}\sum_{u'\in V'}|u\rangle|u'\rangle(a_{uu'}|0\rangle+\sqrt{1-a_{uu'}^2}|1\rangle),$$

$$\frac{1}{\sqrt{nn'}}\sum_{u\in V}\sum_{u'\in V'}|u\rangle|u'\rangle|\bar{\mathbf{\Sigma}}_{uu'}\rangle \rightarrow \frac{1}{\sqrt{nn'}}\sum_{u\in V}\sum_{u'\in V'}|u\rangle|u'\rangle(b_{uu'}|0\rangle+\sqrt{1-b_{uu'}^2}|1\rangle), \tag{21}$$

where $a_{uu'}=\sqrt{\frac{\bar{\mathbf{\Theta}}_{uu'}}{\max_{uu'}(\bar{\mathbf{\Theta}}_{uu'})}}$ and $b_{uu'}=\sqrt{\frac{\bar{\mathbf{\Sigma}}_{uu'}}{\max_{uu'}(\bar{\mathbf{\Sigma}}_{uu'})}}$. We denote this quantum operation as $\mathcal{D}^l, l\in\{1,\ldots,L\}$, where $\mathcal{D}^L$ is used for the quantum readout operation. Similar to the operation $\mathcal{D}^0$, the quantum aggregation transformation can be performed by generating a unitary operator by introducing conditional quantum evolution. Notice that $a_{uu'}$ and $b_{uu'}$ can be viewed as $\sqrt{P_{uu'}}$ in the setting of the single-layer GraphQNTK.

**Quantum readout**. The resulting NTK is embedded into the basis states of a superposition since the algorithm ends up in the fully-connected layers. Similar to the classic readout operation, the summation of all the elements of the NTK matrix at the $L$-th layer is required. We use Conditional Rotation to extract the NTK back to the amplitude, and define a unitary $\mathcal{O}$ which is a generalization of the unitary $\mathcal{U}$, where

$$\mathcal{O}=\sum_{v\in V}\sum_{v'\in V}|v\rangle|v'\rangle\langle v|\langle v'|\otimes\mathcal{D}_{vv'}^L. \tag{22}$$

The unitary $\mathcal{O}$ sums the square root of all the elements of the GraphQNTK matrix. And the resulting GraphQNTK between two input graphs is $\bar{\mathbf{\Theta}}_{GG'}=\frac{(\sum_{u\in V,u'\in V'}\sqrt{\bar{\mathbf{\Theta}}_{uu'}})^2}{n\times n'}$, where $\bar{\mathbf{\Theta}}_{uu'}$ is the GraphQNTK of the last layer.

**Quantum inference to unseen data**. We assume that the test data and the label of the training set are already encoded into the QRAM such that $|\mathbf{k}_*\rangle\in\mathbb{R}^N$, the GraphQNTK between the test graph $G^*$, can be evaluated as the same way to the evaluation between the training data. Let $\bar{\mathbf{\Theta}}\in\mathbb{R}^{N\times N}$ denote the GraphQNTK. The final prediction for a test datapoint $G_*$ is

$$f^*(G^*)=\langle\mathbf{k}_*|\bar{\mathbf{\Theta}}^{-1}|\mathbf{y}\rangle, \tag{23}$$

which requires solving the linear equation $|\mathbf{E}\rangle=\bar{\mathbf{\Theta}}^{-1}|\mathbf{y}\rangle$ and performing inner product estimation on $\langle\mathbf{k}_*|\mathbf{E}\rangle$. A popular quantum algorithm which is designed to solve the quantum linear systems problem (QLSP) is developed by [13], and its runtime is $O(\log(N)\kappa s \text{ polylog}(\kappa s/\epsilon))$ where $s$ is the sparsity of matrix $\mathbf{\Theta}$ and $\kappa$ is the condition number. To realize the quantum speedup, we assume a specific sparsity pattern is created in the quantum storage that only keeps $O(\log N)$ number of non-zero elements of the $N\times N$ GraphQNTK matrix and the well-conditioning is achieved by using Gershgorin circle theorem similar to [71].

## 2.4 Complexity Study

In Sec. 2.3, we discuss how to approximately estimate GNTK using quantum computing paradigm between two input graphs. The time complexity is dominated by the quantum aggregation transformation procedure as it requires encoding the amplitude into an additional register, which takes

Table 1: Classification accuracies on graphs with discrete node attributes. The AttentionGNTK denotes the GNTK with attention mechanism without both sparsity and well conditioning, while the GraphQNTK is the kernel after performing these two transformations to meet the conditions for the use of quantum matrix inversion. The results of other models are taken from [17] except QS-CNN, which we evaluate on our dataset separation.

| Dataset | MUTAG | PROTEINS | PTC | NCI1 | IMDB-B | IMDB-M |
|---|---|---|---|---|---|---|
| WL subtree [57] | $90.4 \pm 5.7$ | $75.0 \pm 3.1$ | $59.9 \pm 4.3$ | $\mathbf{86.0 \pm 1.8}$ | $73.8 \pm 3.9$ | $50.9 \pm 3.8$ |
| AWL [30] | $87.9 \pm 9.8$ | - | - | - | $74.5 \pm 5.9$ | $51.5 \pm 3.6$ |
| RetGK [69] | $90.3 \pm 1.1$ | $75.8 \pm 0.6$ | $62.5 \pm 1.6$ | $84.5 \pm 0.2$ | $71.9 \pm 1.0$ | $47.7 \pm 0.3$ |
| GNTK [17] | $90.0 \pm 8.5$ | $75.6 \pm 4.2$ | $\mathbf{67.9 \pm 6.9}$ | $84.2 \pm 1.5$ | $76.9 \pm 3.6$ | $52.8 \pm 4.6$ |
| GCN [39] | $85.6 \pm 5.8$ | $76.0 \pm 3.2$ | $64.2 \pm 4.3$ | $80.2 \pm 2.0$ | $74.0 \pm 3.4$ | $51.9 \pm 3.8$ |
| GraphSAGE [21] | $85.1 \pm 7.6$ | $75.9 \pm 3.2$ | $63.9 \pm 7.7$ | $77.7 \pm 1.5$ | $72.3 \pm 5.3$ | $50.9 \pm 2.2$ |
| PatchySAN [49] | $92.6 \pm 4.2$ | $75.9 \pm 2.8$ | $60.0 \pm 4.8$ | $78.6 \pm 1.9$ | $71.0 \pm 2.2$ | $45.2 \pm 2.8$ |
| GIN [67] | $89.4 \pm 5.6$ | $76.2 \pm 2.8$ | $64.6 \pm 7.0$ | $82.7 \pm 1.7$ | $75.1 \pm 5.1$ | $52.3 \pm 2.8$ |
| QS-CNN [70] | $\mathbf{93.1 \pm 4.7}$ | $\mathbf{78.2 \pm 4.6}$ | $66.0 \pm 4.4$ | $81.4 \pm 2.6$ | $72.1 \pm 3.7$ | $46.2 \pm 4.2$ |
| AttentionGNTK | $90.0 \pm 8.5$ | $76.2 \pm 3.8$ | $66.2 \pm 5.1$ | $84.1 \pm 1.2$ | $\mathbf{76.9 \pm 3.2}$ | $\mathbf{52.9 \pm 3.5}$ |
| GraphQNTK | $88.4 \pm 6.5$ | $71.1 \pm 3.2$ | $62.9 \pm 5.0$ | $77.2 \pm 2.7$ | $73.3 \pm 3.6$ | $48.1 \pm 4.3$ |

$O(\log(nd)log(1/\Delta)/\epsilon)$ time. Other quantum operations including estimation of the inner product, estimation of the GNTK within the neighborhood aggregation and the fully-connected feature updating and quantum readout are totally unitary operations which can be efficiently performed under the regime of quantum computing. For estimating GNTK of each pairs of the graphs $(G, G')$ where $G, G' \in \mathcal{G}$, each element of GraphQNTK $\bar{\Theta}$ can be generated simultaneously by introducing auxiliary index registers. The quantum runtime is $O(\log(Nnd))$. However, evaluating GNTK of the infinite-width-limit attention requires computing the kernel where the input is two same graphs, which can be implemented in time $O(N)$. The result should be stored in QRAM in advance which will be used to update GNTK corresponding the multi-head attention as described in Eq. 14. Therefore, it takes $O(N \log(Nnd))$ time to train the proposed quantum graph learning model, which achieves quadratic speedup compared to the existing GKs and completed approaches with $O(N^2)$ time.

## 3 Experiments

We evaluate our method for both GNTK and GraphQNTK with attention mechanism on several graph classification datasets involving either discrete or continuous attributes. All the experiments are performed on a workstation with a single machine with 1TB memory, one physical CPU with 28 cores Intel(R) Xeon(R) W-3175X CPU @ 3.10GHz, and a single GPU (Nvidia Quadro RTX 8000). For our method and all the compared models, We follow the same setting as [17, 67], and report the average test accuracy and its standard deviation over a 10-fold cross validation on each dataset.

### 3.1 Experiments Setup

**Datasets.** For graph with discrete attributes, the benchmark datasets include four bioinformatics datasets MUTAG, PTC, NCI1, PROTEINS and three social network datasets IMDB-BINARY, IMDB-MULTI. For each graph, the input attributes is category of the node and they are transformed to one-hot encoding representations. For datasets where the graphs have no node features, i.e. only graph structure matters, we use degrees as input node features. For graph with continuous attributes, we selcect four benchmark datasets including ENZYMES, PROTEINS full, BZR, COX2. All the datasets can be found in [38]. The statistic information of the datasets are given in Tab. 3 in Appendix.

**Compared baselines.** We compare our method with state-of-the-art GKs such as WL kernel [57], AWL [30], RetGK [69], GNTK [17], WWL [62], and GNNs including GCN [39], PatchySAN [49], GCKN [10], GraphSAGE [21] and GIN [67]. For quantum graph learning, there are very few baseline available. We report the performance of the quantum walk based subgraph convolutional neural network (QS-CNN) developed by [70]. The data separation we use is the same as [67] for graph

Table 2: Classification accuracies on graphs with continuous attributes. The accuracies of other models are taken from [10]. We only take the results of GCKN under the supervised learning for a fail comparison. We utilize the similar settings that preprocess the continuous node features to a normalized feature vector as in [62] for fair comparison (Note that the data encoded into the QRAM requires normalization, thus it is reasonable to use this data-prepossessing operation).

| Dataset | ENZYMES | PROTEINS | BZR | COX2 |
|---|---|---|---|---|
| RBF-WL [62] | $68.4 \pm 1.5$ | $75.4 \pm 0.3$ | $81.0 \pm 1.7$ | $75.5 \pm 1.5$ |
| HGK-WL [47] | $63.0 \pm 0.7$ | $75.9 \pm 0.2$ | $78.6 \pm 0.6$ | $78.1 \pm 0.5$ |
| HGK-SP [47] | $66.4 \pm 0.4$ | $75.8 \pm 0.2$ | $76.4 \pm 0.7$ | $72.6 \pm 1.2$ |
| WWL [62] | $\mathbf{73.3 \pm 0.9}$ | $\mathbf{77.9 \pm 0.8}$ | $84.4 \pm 2.0$ | $78.3 \pm 0.5$ |
| GNTK [17] | $69.6 \pm 0.9$ | $75.7 \pm 0.2$ | $85.5 \pm 0.8$ | $79.6 \pm 0.4$ |
| GCKN [10] | $72.8 \pm 1.0$ | $77.6 \pm 0.4$ | $86.4 \pm 0.5$ | $81.7 \pm 0.7$ |
| AttentionGNTK | $69.2 \pm 1.1$ | $76.8 \pm 1.2$ | $\mathbf{86.7 \pm 1.3}$ | $\mathbf{82.1 \pm 0.4}$ |
| GraphQNTK | $64.8 \pm 0.7$ | $72.5 \pm 0.3$ | $80.1 \pm 1.7$ | $74.3 \pm 1.9$ |

datasets with discrete attributes. For graph dataset with continuous attributes, we follow the same protocol as used in [62] to normalize the input feature vectors for a fair comparison.

## 3.2 Experiment Results

We apply different hyper-parameter settings to $L \in \{2, 4, 6, 8\}$ and $R \in \{1, 2, 3\}$ and select the model with the best averaged accuracy. We test the kernel regression using SVM classifier and the regularization parameter is determined using the search protocol which is the same as the [17]. We report the performance of the quantum approximate GNTK before and after the matrix sparsity and conditioning operations. The numerical results are listed in Tab. 1 for datasets with discrete attributes and Tab. 2 for datasets with continuous attributes. The attention method we integrate to the infinite-width GNNs brings to an improvement in the performance of the model. The results show that the GNTK with attention mechanism achieves better classification accuracy for graph data with medium number of nodes and edges. It is demonstrated that the infinite-width-limit attention captures global node similarity semantics and learns effective structure of the provided graph, which brings an remarkable accuracy improvement of the model compared with the vanilla GNTK [17]. Moreover, our model performs better than QS-CNN on more than 60% of the datasets with discrete attributes, given the caveat that QS-CNN is a hybrid graph learning model where the contribution of the classic components (CNNs, spatial message passing) in their model cannot be ignored. While the matrix sparsity and conditioning operations have a great influence on the model's performance, it can be found that the classification performance of GNTK evaluated by quantum algorithms is still comparable with that of GKs and vanilla GNNs, where a tradeoff exists between the performance of the model and the quantum computational efficiency.

## 4 Conclusion and Broader Impact

This paper has presented a quantum graph learning model to characterize the structural information while using quantum parallelism to improve computing efficiency. We propose quantum algorithm to approximately estimate the neural tangent kernel of the underlying graph neural network where a multi-head quantum attention mechanism is introduced to incorporate semantic similarity of nodes. Empirical results on graph classification tasks as well as theoretical analysis show the superiority of our method. The limitation of the paper is that currently it only addresses graph-level embedding and we leave node-level quantum learning for future work. Our work may raise concerns for encryption, privacy protection etc. when the quantum hardware become more feasible.

## Acknowledgement

This work was partly supported by National Key Research and Development Program of China (2020AAA0107600), National Natural Science Foundation of China (61972250, 72061127003), and Shanghai Municipal Science and Technology (Major) Project (2021SHZDZX0102, 22511105100).

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
