# OpenReview forum: "GraphQNTK: Quantum Neural Tangent Kernel for Graph Data"
_NeurIPS.cc/2022/Conference — NeurIPS 2022 Accept_

### Official Review · Reviewer_7X59 · 2022-07-10

**Rating:** 5
**Confidence:** 3
**Soundness:** 4 excellent
**Presentation:** 3 good
**Contribution:** 3 good

**Summary:**

The paper introduces an attention based graph neural tangent kernel and proposes a quantum algorithm to approximate it efficiently. Experiments are conducted on seven datasets with discrete attributes and four datasets with continuous attributes.

**Questions:**

In addition to reducing the runtime complexity, what are the motivations of the proposed model?

**Limitations:**

Yes, the authors have addressed these in their work.

**Strengths And Weaknesses:**

Strength:

1. With the attention mechanism, the proposed quantum graph learning model is better at capturing global information of the graph.
2. The runtime complexity of the quantum algorithm can be reduced from $O(N^2)$, which appears to normal graph kernel neural network, to $O(N)$.

Weaknesses:

1. From the reported results, the performance of the proposed GraphQNTK is not competitive compared to the baselines.
2. The ability of capturing global information is advertised as a strength for the proposed model. However, no further theoretical argument or numerical experiment in the main body is presented to justify this feature.

---

> ### Author Response · Authors · 2022-08-02
> **Response to Reviewer 7X59 Part 2**
>
> **Q3: In addition to reducing the runtime complexity, what are the motivations of the proposed model?**
>
> **A3:** Thanks for your question. It is indeed one of our motivations to reduce the runtime complexity of deep GNNs benefited from quantum parallelism. Additionally, we restate our motivations as follows. Notice that these points actually correspond to the three contributions in the paper, and we re-sort out their logic and clarify them more clearly.
> 1. Attention-enhanced GNTK. Infinite-width GNN is a well established term in the GNN literature [a,b], and GNTK [a] is a powerful tool to analyze the GNN. However, the traditional feature aggregation of the vanilla GNTK is straightforward, limiting itself within joint neighborhoods. In this paper, a multi-head attention mechanism is introduced to properly incorporate semantic similarity information of disconnected nodes but with similar features, which is demonstrated in supplementary experiments in our rebuttal A2.
> 2. Kernel methods for evaluating the dynamics of wide and deep GNNs. It is generally hard to train a deep GNN with attention, especially when the width of GNN, the width of attention layer, or the number of heads goes to large. We properly incorporate infinite-width multi-head attention into GNTK by using NTK theory. We use kernel methods to capture the dynamics of infinite-width GNN (Section 2.1) with infinite-width attention (Section 2.2), thus avoiding the huge overhead of training a wide and deep GNN.
> 3. Speedup introduced by quantum computing. Although GNTK is a useful method to train an infinite-width GNN, its cost still grows quadratically with respect to the volume of data, which is intractable for large datasets. We re-design the Attention-enhanced GNTK by splitting it into small components and reuniting them using the quantum linear algebra subroutines. The produced quantum graph kernel – GraphQNTK, theoretically reduces the computational complexity to $O(N)$ benefited from the quantum parallelism.
>
> [a] Du S S, Hou K, Salakhutdinov R R, et al. Graph neural tangent kernel: Fusing graph neural networks with graph kernels. NeurIPS, 2019.
>
> [b] Huang W, Li Y, Du W, et al. Towards Deepening Graph Neural Networks: A GNTK-based Optimization Perspective. ICLR, 2022.

---

> ### Author Response · Authors · 2022-08-02
> **Response to Reviewer 7X59 Part 1**
>
> Thank you for your insightful comments! We hope our answers below address all your concerns.
>
> **Q1: From the reported results, the performance of the proposed GraphQNTK is not competitive compared to the baselines.**
>
> **A1:** Thanks for your valuable question. Please refer to A4 of our rebuttal to Reviewer 4zaS for a detailed explanation. The main reason is that there is a tradeoff between the quantum acceleration and the model’s performance. To satisfy the prerequisites for quantum speedup and better classically reproduce the process of computing the quantum graph kernel--GraphQNTK in real quantum devices, we introduce matrix sparsity and well-conditioned operations acting on the intermediate kernel matrix. The consequence is that the classification accuracy of GraphQNTK decreases by 3% to 7% for different datasets. It is noteworthy that GraphQNTK achieves a quadratic speedup compared to the existing graph kernel methods. We believe that the strategy suggested in this paper will lead to a new quantum machine learning breakthrough and better exploit the immense benefits of quantum computing when quantum devices are consistently available.
>
> **Q2: The ability of capturing global information is advertised as a strength for the proposed model. However, no further theoretical argument or numerical experiment in the main body is presented to justify this feature.**
>
> **A2:** Thanks for your insightful comments and we will clarify the global information perception we claimed. From the theoretical perspective, the transformer (Eq.12) captured the semantic information between each pair of (connected and disconnected) nodes with similar features. Consider $\mathbf{G}=\frac{\mathbf{Q} \mathbf{K}^{\top}}{\sqrt{s}}$ in Eq.10, where $\mathbf{Q}$ and $\mathbf{K}$ are linear transformation of the node feature matrix and we ignore the superscript and the subscript for simplicity. The $\mathbf{G}$ can be viewed as a matrix whose element corresponds to the similarity of each pair of nodes. Then the operation $\widehat{\mathbf{H}}=\zeta\left(\mathbf{G}\right) \mathbf{V}$ transforms the node features of the last layer to the next layer depending on the node similarity. This enables the model to make better use of the graph structure to transmit information and perceive topology information over long distances. In order to demonstrate it empirically, we compare the performance between GNTK and our model at the same number of layers. The number in parentheses in the following table indicates the number of layers.
>
>
> |          | GNTK(4)         | Ours(4)        | GNTK(6)         | Ours(6)        | GNTK(8)         | Ours(8)        |
> | -------- | -------------- | -------------- | -------------- | -------------- | -------------- | -------------- |
> | PTC      | 62.9 $\pm$ 7.2 | 64.9 $\pm$ 5.3 | 63.5 $\pm$ 6.8 | **66.2 $\pm$ 5.1** | 65.2 $\pm$ 7.9 | 63.4 $\pm$ 6.6 |
> | NCI1     | 83.6 $\pm$ 2.1 | **84.1 $\pm$ 1.2** | 84.0 $\pm$ 0.9 | 83.8 $\pm$ 1.2 | 82.9 $\pm$ 1.8 | 82.3 $\pm$ 2.2 |
>
>
> The numerical results demonstrate that our model (attentionGNTK)  reaches the peak of classification accuracy when the number of layers is small, while GNTK needs more layers to reach, indicating that our model is easier to capture the global structure information of the graph.

---

> ### Author Response · Authors · 2022-08-07
> **We would love to hear your feedback on our rebuttal**
>
> Dear Reviewer 7X59,
>
> Thanks again for your review. We hope our answers could increase your confidence. As the discussion period is close to the end and we have not yet heard back from you, we would be glad to see if our rebuttal response has addressed your concerns questions/concerns.
>
> We are more than happy to discuss further if you have any further concerns and issues, please kindly let us know your feedback. Thank you for your time and help!

---

> > ### Comment · Reviewer_7X59 · 2022-08-09
> > **Thanks for the response**
> >
> > Thanks for the authors’ responses. The additional experiments and explanations addressed my questions and concerns. I will raise my score to 5 accordingly.

---

### Official Review · Reviewer_4zaS · 2022-07-11

**Rating:** 7
**Confidence:** 4
**Soundness:** 3 good
**Presentation:** 3 good
**Contribution:** 4 excellent

**Summary:**

Graph neural tangent kernel (GNTK) is a powerful tool to combine the expressive ability of Graph Neural Networks (GNNs) and the trainability of Graph Kernels (GKs). The authors argue that, the vanilla feature aggregation operation in existing GNTK is too simple to capture diverse graph properties and training GNTK requires substantial computing efforts due to the kernel complexity.

To overcome these problems, the authors design a novel quantum graph learning model for graph classification. On one hand, to improve GNTK’s expressiveness, the multi-head attention mechanism is for the first time integrated into GNTK. On the other hand, the attention-enhanced GNTK is re-designed and re-implemented with a quantum version for improving the training speed. This is achieved by introducing the quantum aggregation transformation algorithm and the quantum kernel estimation algorithm. These two quantum algorithms make use of quantum superposition and quantum parallelism to realize the transformation of graph tangent kernel matrix, while preserving the dynamics of GNNs with infinite-width graph convolution and attention layers.

Experiments show the proposed quantum model performs competitively against both GNNs and the GKs, and theoretical analysis explains the speedup benefited from quantum algorithms.


**Questions:**

1. It seems that only the node feature is encoded into the proposed quantum graph learning model before the training starts. How is the structural information of the graph integrated into the whole training process?

2. In the experiment, the performance of GraphQNTK is significantly weaker than that of AttentionGNTK. Could you explain this phenomenon?

3. I am fine with the simulation on your classic computer. But what is the exact cost for your training? Because this is related to the cost for verifying your model’s advantage on a classic computer.


**Limitations:**

Please see my questions.

**Strengths And Weaknesses:**

Strengths:
+ This work is clearly motivated and well written
The background of the research, the motivation of quantum graph learning for graph classification and the related work are all clearly stated and summarized.
+ The first for introducing attention mechanism in quantum graph learning
The multi-head attention mechanism is introduced into the GNTK by adapting the GNTK with reasonable modification by the incorporation of more node’s semantic similarity between the adjoining neighbors and nodes with similar features. The formulation of the resulting quantum version of GNTK is achieved by the association of quantum linear algebra subroutines.
+ Quantum speedup for improved GNTK
To employ the natural properties of quantum computing such as superposition and parallelism, the quantum aggregation transformation algorithm and the quantum kernel estimation algorithm are designed for efficiently calculation of the GNTK. The theoretical analysis demonstrates that training the quantum-based GNTK is quadratically faster than its classical analogue.
+ Extensive numerical evaluation
This paper demonstrates the power of the quantum GNTK model on several graph classification datasets, achieving competitive performance with respect to the GNNs and the GKs.

Weaknesses:
- The quantum algorithm described in this work is a quantum reconstruction of GNTK with extra attention layers. However, the whole training process is the kernel method, and the relationship between the proposed model and the GNNs is very weak.
- Although computing GNTK inside quantum devices using the proposed quantum algorithms is very efficient after encoding the node attributes and compiling the specified quantum circuit. It seems that optimization of the whole model requires multiple execution of the same circuit and frequent interactions between classical and quantum environments, which could introduce additional expenses.

Overall, I think the paper is technically novel and considering there is no easy way to deploy the method in a real quantum machine due to hardware limitation in near future, I am fine with the current experiments. I vote to accept this work.

---

> ### Author Response · Authors · 2022-08-02
> **Response to Reviewer 4zaS Part 3**
>
> **Q4: In the experiment, the performance of GraphQNTK is significantly weaker than that of AttentionGNTK. Could you explain this phenomenon?**
>
> **A4:** Thanks for your valuable question. Indeed, the graph classification accuracy of AttentionGNTK is stronger than that of GraphQNTK (a decrease of 3% to 7% for different datasets). Maybe we can find the answer by reaffirming the relationship and differences between these two models.
>
> GNTK is the kernel metric measuring the similarity between two input graphs. To properly incorporate semantic similarity information of nodes into the model, the infinite-width multi-head attention is introduced and the result enhanced kernel -- AttentionGNTK is defined in Eq.15. However, the runtime of measuring each pair of graphs among the datasets is $O(N^2)$, which is intractable for large datasets. Inspired by the simultaneous amplitude transformation induced by the quantum superposition and estimating the DNN's output via quantum parallelism [a], we re-design the AttentionGNTK by splitting it into small components and reuniting them using the quantum linear algebra subroutines. The produced quantum graph kernel -- GraphQNTK, theoretically reduces the computational complexity to $O(N)$ benefited from the quantum parallelism. However, just as there is no such thing as a free lunch, there is a tradeoff between the quantum acceleration and the model's performance. To fulfill the applicable condition to which the quantum subroutines can be adaptive and achieve quantum speedup, the GraphQNTK needs to be sparse and well-conditioned. To overcome these problems, we adopt the idea in [a] which assumes a specific sparsity pattern in the quantum storage and uses the Gershgorin circle theorem.
>
> In conclusion, AttentionGNTK is an enhanced graph similarity metric compared with GNTK. The numerical results in Tab.1 and Tab.2 show that AttentionGNTK outperforms GNTK on 7 out of 10 datasets, reflecting that the introduced infinite-width multi-head attention is useful and can better capture distinct properties between different graphs. Furthermore, we compare our model with another quantum-inspired graph learner QS-CNN [b]. Our model surpasses QS-CNN on 4 out of 6 datasets and performs similarly on the rest 2 datasets. It shows the superiority of our model in quantum graph learning. Moreover, it is noticed that there are a large number of classical layers in QS-CNN, which greatly weakens its overall value. As a consequence of matrix sparsity and well-conditioned operations, the classification accuracy of GraphQNTK decreases by 3% to 7% for different datasets. At the cost of acceptable loss of classification accuracy, it is worth mentioning that GraphQNTK achieves a quadratic speedup compared to the existing graph kernel methods. We believe that the strategy suggested in this paper will lead to a new quantum machine learning breakthrough and better exploit the immense benefits of quantum computing when quantum devices are consistently available.
>
> [a] Zlokapa A, Neven H, Lloyd S. A quantum algorithm for training wide and deep classical neural networks. arXiv:2107.09200, 2021.
>
> [b] Zhang Z, Chen D, Wang J, et al. Quantum-based subgraph convolutional neural networks. Pattern Recognition, 2019.
>
> **Q5: I am fine with the simulation on your classic computer. But what is the exact cost for your training? Because this is related to the cost for verifying your model’s advantage on a classic computer.**
>
> **A5:** Thanks for your valuable suggestion. We supplement the training time of our model on four selected datasets and compare it with two other models. To make a fair comparison, we set the layers of all the models to 2. All the experiments are performed on a workstation with a single machine with 1TB memory, one physical CPU with 28 cores Intel(R) Xeon(R) W-3175X CPU @ 3.10GHz, and a single GPU (Nvidia Quadro RTX 8000).
>
> |      | MUTAG | NCI1  | IMDB-B | IMDB-M |
> | ---- | ----- | ----- | ------ | ------ |
> | Ours | 14sec | 21min | 4min   | 9min   |
> | GIN  | 22sec | 67min | 19min  | 24min  |
> | GNTK | 9sec  | 18min | 4min   | 7min   |
>
> Although the speedup introduced by the quantum algorithm depends on the quantum devices, it shows that our proposed model still has a computational overhead reduction when training on classic computers. The running time is slightly higher than that of GNTK which is a lack of attention mechanism. It is noticed that our model. The runtime of our model is apparently faster than that of GIN.
>
> It is worth mentioning that the quadratic quantum speedup will be realized when the quantum hardware becomes more feasible.

---

> ### Author Response · Authors · 2022-08-02
> **Response to Reviewer 4zaS Part 2**
>
> **Q3: It seems that only the node feature is encoded into the proposed quantum graph learning model before the training starts. How is the structural information of the graph integrated into the whole training process?**
>
> **A3:** Thanks for your insightful comment which is worth careful discussion in our revised version as well as in our response. Unlike regular data, such as image and text, the structural information of graph data is of great importance for analyzing the underlying data patterns. Many classic graph learners generalize the convolution and pooling operations to incorporate the structural information for graph data mining [a].
>
> Quantum machine learning has recently drawn great attention, aiming at accelerating the training of traditional models and finding useful patterns of data [b]. However, quantum computing for graph learning is still in its early stages [c]. **As far as we know, no quantum graph learning model can both effectively describe the structural information and naturally utilize the advantage of quantum computing, i.e., quantum parallelism to process the graph data**.
>
> To characterize the graph structure, we design two novel quantum algorithms namely Quantum Aggregation Transformation [line 225 to line 248] and Quantum Kernel Estimation [line 249 to line 271]. The former is responsible for aggregating the adjoint neighbor's feature in accordance with the input graph structure. While the latter captures the multi-hop semantic information between disconnected nodes with similar features. These two algorithms are developed using the unitary operation which is naturally adaptive to the evolution of quantum systems [d]. Together with the node feature encoded into the quantum states, these two quantum algorithms can efficiently process graph-structure data and retain the acceleration effect of quantum computing.
>
> [a] Wu Z, Pan S, Chen F, et al. A comprehensive survey on graph neural networks. TNNLS, 2020.
>
> [b] Huang H Y, Broughton M, Mohseni M, et al. Power of data in quantum machine learning. Nature communications, 2021.
>
> [c] Tang Y, Yan J, Edwin H. From Quantum Graph Computing to Quantum Graph Learning: A Survey. arXiv:2202.09506, 2022.
>
> [d] Nielsen M A, Chuang I. Quantum computation and quantum information. 2002.

---

> ### Author Response · Authors · 2022-08-02
> **Response to Reviewer 4zaS Part 1**
>
> Thank you for your constructive comments! We hope our answers below address all your concerns.
>
> **Q1: The quantum algorithm described in this work is a quantum reconstruction of GNTK with extra attention layers. However, the whole training process is the kernel method, and the relationship between the proposed model and the GNNs is very weak.**
>
> **A1:** Thanks for your attention to our work. Basically speaking, our model is designed analog to the infinite-width GNN with attention mechanism whose dynamics is captured by the Quantum Graph Neural Tangent Kernel (GraphQNTK). The connection between the construction of GraphQNTK and the architecture of GNN with attention is illustrated in Fig. 2 in the paper. We will give a more intuitive explanation here.
> 1. The relation between GNN and GNTK.
> The dynamics of training the GNTK is equivalent to training an infinitely-wide GNN initialized with random weights trained with gradient descent. In Section 2.1, we give the mathematical formulation of GNTK, where Eq. 5 and Eq. 8 correspond to the feature aggregation of adjoint neighbors (Eq. 1) and feature update of the central node (Eq. 2), respectively.
> 2. The relation between GNN with attention mechanism and GraphQNTK.
> To facilitate the capability of information aggregation within the disconnected nodes with similar features, multi-head attention is adopted in our model. We transform the infinite-width multi-head attention to a novel neural tangent kernel, as discussed in Eq. 12 and Eq. 13. Training this kernel is equivalent to training the multi-head attention layer whose output dimension and the number of heads go to infinity. Finally, this kernel is integrated into GNTK and generate the GraphQNTK (ignoring the sparsity and well-conditioning operations).
>
> Thus, the link between the proposed model and the GNNs is strong as GraphQNTK is derived from the functions of GNNs.
>
> **Q2: It seems that optimization of the whole model requires multiple execution of the same circuit and frequent interactions between classical and quantum environments, which could introduce additional expenses.**
>
> **A2:** Thanks for your question. This is indeed a standing problem that the interaction (such as optimization of parametrized quantum gates, cascading of classical and quantum layers) between classical and quantum environments may bring up huge computational overheads [a]. When designing quantum algorithms, the interaction between the two environments must be taken into account carefully, otherwise, the potential quantum acceleration may diminish.
>
> However, our model successfully bypasses this obstacle, since the GraphQNTK can be calculated using the quantum devices without the help of classical systems. Concretely, we explain it from three aspects: input, training and readout. First, encoding data into QRAM only occurs a single time before training starts. Second, no need for a classic optimizer to optimize variational quantum circuits because no adjustable arguments exist after model initialization. Third, the readout from the quantum output can be efficiently performed by the quantum inner product estimation (Eq.23). Therefore, the additional expenses introduced by the interactions between the two different systems can be neglected, which further reflects the acceleration effect of our quantum model. We believe this is a big advantage of our method compared with many so-called quantum machine learning models which consider little about the interaction costs between classic and quantum environments.
>
> [a] Cerezo M, Arrasmith A, Babbush R, et al. Variational quantum algorithms. Nature Reviews Physics, 2021.

---

> ### Author Response · Authors · 2022-08-07
> **We would love to hear your feedback on our rebuttal**
>
> Dear Reviewer 4zaS,
>
> Thanks again for your review. We hope our answers could increase your confidence. As the discussion period is close to the end and we have not yet heard back from you, we would be glad to see if our rebuttal response has addressed your concerns questions/concerns.
>
> We are more than happy to discuss further if you have any further concerns and issues, please kindly let us know your feedback. Thank you for your time and help!

---

> ### Comment · Reviewer_4zaS · 2022-08-10
> **Reply to the author's response**
>
> I appreciate the detiled feedback from authors. I have read the replies and other reviews.  All my concerns have been fully addressed, and I have no more questions. I incline to accept this paper, so I would like to increase my score to 7.

---

### Official Review · Reviewer_DbrA · 2022-07-11

**Rating:** 4
**Confidence:** 2
**Soundness:** 3 good
**Presentation:** 2 fair
**Contribution:** 2 fair

**Summary:**

The authors propose a Quantum Neural Tangent Kernels (QNTK) approach for GNNs.
The QNTK is based on GNTK, which is briefly covered by the authors, and QNTK is supposed to offer an accelerated version of GNTK through quantum computing.
The authors suggest to incorporate an attention mechanism (a transformer) in order to capture long range interaction of the graph data (as opposed to only 1-hop ring propagation in GCN) to learn global information about the graph.

The authors examine the proposed method on the graph classification task, on several datasets, where the achieved accuracy is less than considered or other existing methods.

**Questions:**

1. Please clarify what do you mean by infinite width of GNNs, and what is the actual width of GNNs.
2.Please demonstrate your method with more layers.
3.Please provide runtimes for your method and compare it to other methods.
4.Can you please explain why your method is not working on node classification?

**Limitations:**

The authors discussed potential negative societal impact adequately .

**Strengths And Weaknesses:**

Strengths:
1.The authors propose a method that involves quantum computing and GNNs, a subject that is relatively new and less researched.
2.The proposed method is claimed to accelerate GNNs - this is a major point in the field of GNNs.

Weaknesses:
1. From an originality point of view, the concept of QNTK was already proposed in different papers that were not considered in the paper (see for example "Representation Learning via Quantum Neural Tangent Kernels" (arXiv:2111.04225) and "Quantum tangent kernel (arXiv:2111.02951)).

2.It is not clear what do the authors mean by an infinite width GNN. In the end, the method needs to be deployed in code, so does it mean that the number of channels is simply very large? It is not provided in text what the actual width of the network is.

3.From lines 75-76 it can be thought that the authors propose to use GAT, and thus the claim that it is non-local as opposed to the standard propagation of GCN is not in order. I suggest that the authors clarify that their suggestion is to use a transformer.

4.The authors claim a computational cost reduction from O(N^2) to O(N). However, when deploying a transformer that considers all pairs of nodes, it is not clear to me how the costs are saved.

5.Figure 1 is not self contained and also not explained in text properly. It is not clear what 'SVC' and 'SVR' means.

6.Line 108 is not clear. The authors need to define what k* means. What is x* ?

7.The experimental results show inferior results compared to the considered methods.

8.The authors claim for a significant acceleration of computation. However, it is not provided in the experiments. I think that the authors should present training and inference time to compare.

9.The authors consider a few number of layers, up to 8 layers. It is known that many GNNs tend to oversmooth as more layers are added. It is interesting to know how the network behaves with a more layers like 32 and 64.

10.The authors state that the current paper is focused on graph classification and not node classification. However, it is not clear what is the reason for this limitation. Is  it a limitation of the method?

---

> ### Author Response · Authors · 2022-08-02
> **Response to Reviewer DbrA Part 3**
>
> **Q6: Line 108 is not clear. The authors need to define what $k\star$ means. What is $x\star$ ?**
>
> **A6:** Thanks for your question. $x*$ denotes the input graph in the test set. Vector $k*$ is the NTK between $x*$ and each graph in the training set. We have revised it in the newly submitted version.
>
> **Q7: The experimental results show inferior results compared to the considered methods.**
>
> **A7:** Thanks for your question. It is in deed of great difficulty for native quantum machine learning to beat the existing classical deep learning methods in terms of classification accuracy. This is mainly due to the fact that quantum machine learning is in its infancy and most quantum algorithms can not be used as flexibly as classical algorithms. The main reason why GraphQNTK is weaker than AttentionGNTK is that there is a tradeoff between the quantum speedup and the model's performance, e.g., matrix sparsity and well-conditioned operations. Please refer to A4 of our rebuttal to Reviewer 4zaS for a more detailed explanation.
>
> **Q8: The authors claim for a significant acceleration of computation. However, it is not provided in the experiments. I think that the authors should present training and inference time to compare.**
>
> **A8:** Thanks for your comment. Please refer to A5 of our rebuttal to Reviewer 4zaS for a detailed explanation. Here we briefly give an illustration. The quadratic speedup of our proposed algorithm is provided by quantum parallelism, which depends on quantum physical devices. To validate the potential acceleration, we conduct a simulation on a classic computer and the rusults show that our model still has advantages in terms of running time compared with deep graph learning models such as GIN. Our model's running time is longer than GNTK's, which lacks an attention mechanism. However, the acceleration capability will be unlocked when our quantum algorithms run on quantum physical devices.
>
> **Q9: The authors consider a few number of layers, up to 8 layers. It is known that many GNNs tend to oversmooth as more layers are added. It is interesting to know how the network behaves with more layers like 32 and 64.**
>
> **A9:** Thanks for your suggestion. When we figure out the effect of different number of layers on model performance, we do try to increase the number of layers to more than 10. However, we find that on all datasets, the classification accuracy of the model on the test data will decrease significantly. Thus we think the oversmooth still exists in the proposed model. But it is noticed that our model is more resilient to the addition of more layers compared with GIN where the number of layers is generally limited to less than 5. We compare the graph classification accuracy bewteen GIN and our model at the same number of layers. The number in parentheses in the following table indicates the number of layers. The results are as follows.
>
> |          | GIN(4)         | Ours(4)        | GIN(6)         | Ours(6)        | GIN(8)         | Ours(8)        |
> | -------- | -------------- | -------------- | -------------- | -------------- | -------------- | -------------- |
> | MUTAG    | 87.6 $\pm$ 6.2 | 89.1 $\pm$ 7.8 | 88.5 $\pm$ 5.6 | 90.0 $\pm$ 8.5 | 86.2 $\pm$ 6.4 | 88.4 $\pm$ 7.4 |
> | PROTEINS | 75.5 $\pm$ 3.0 | 75.0 $\pm$ 4.1 | 74.3 $\pm$ 3.0 | 76.1 $\pm$ 3.8 | 72.8 $\pm$ 3.5 | 74.2 $\pm$ 4.4 |
> | PTC      | 62.8 $\pm$ 5.0 | 64.9 $\pm$ 5.3 | 62.0 $\pm$ 6.2 | 66.2 $\pm$ 5.1 | 61.2 $\pm$ 7.1 | 63.4 $\pm$ 6.6 |
> | NCI1     | 82.3 $\pm$ 3.6 | 84.1 $\pm$ 1.2 | 80.1 $\pm$ 2.4 | 83.8 $\pm$ 1.2 | 77.2 $\pm$ 3.3 | 82.3 $\pm$ 2.2 |
> | IMDB-B   | 73.2 $\pm$ 4.1 | 75.7 $\pm$ 2.8 | 74.4 $\pm$ 6.0 | 76.9 $\pm$ 4.3 | 72.1 $\pm$ 5.2 | 75.1 $\pm$ 4.0 |
> | IMDB-M   | 51.7 $\pm$ 3.7 | 52.0 $\pm$ 4.1 | 52.0 $\pm$ 2.6 | 51.9 $\pm$ 3.7 | 48.2 $\pm$ 4.3 | 50.3 $\pm$ 4.5 |
>
> **It is shown that our model (AttentionGNTK) is more robust compared with GIN when the number of layers becomes larger**. The main reason is that an additional feature aggregation, e.g., the transformer, can slow down the convergence rate, which is consistent with the observations in [a] that connectivity enhancement can help wide and deep GNNs to avoid a discrepancy between prediction and the ground truth.
>
> [a] Huang W, Li Y, Du W, et al. Towards Deepening Graph Neural Networks: A GNTK-based Optimization Perspective. ICLR, 2022.
>
> **Q10: The authors state that the current paper is focused on graph classification and not node classification. However, it is not clear what is the reason for this limitation. Is it a limitation of the method?**
>
> **A10:** Thanks for your question. Indeed, so far our model can only handle the graph classification tasks because it can be attributed to the graph kernel method, which is specially designed to measure the similarity between input graphs. We leave node-level quantum learning for future work.

---

> > ### Comment · Reviewer_DbrA · 2022-08-09
> > **An answer to the authors**
> >
> > Dear authors,
> > Thank you for your rebuttal.
> >
> > Below are my issues that still stand after reading your rebuttal:
> >
> > A2: I appreciate the detailed answer. In the end I am still confused about what exactly the infinite width is, and it is not explained in the main paper. I think that it should be.
> >
> > A3-A6: I understand that this is an approach that is supposed to be deployed at a later stage when quantum devices are more achievable, but I can not fully understand the potential without a major revision of the text. Future potential is important but it needs to be very clear from the text.
> >
> > I thus keep my score as is.
> >
> > With kind regards,
> >
> > Reviewer DbrA

---

> > > ### Author Response · Authors · 2022-08-09
> > > **Response to Reviewer DbrA Part 2**
> > >
> > > **Q2: I understand that this is an approach that is supposed to be deployed at a later stage when quantum devices are more achievable, but I can not fully understand the potential without a major revision of the text. Future potential is important but it needs to be very clear from the text.**
> > >
> > > **A2:** Thanks for your comment. It is true that the quantum speedup (from $O(N^2)$ to $O(N)$) to train a graph deep learning model benefited from quantum parallelism depends on the available quantum devices, and we appreciate that this contribution of our paper is acknowledged. In the Section 2.4, we systematically investigate the complexity of our model. Specifically, the speedup ratio of each quantum algorithm is analyzed one by one when re-design the attention-enhanced GNTK using quantum linear algebra subroutines. It is worth noting that our model still saves remarkable running time when training on a classical computer compared with GIN (see Table 4). We have revised the paper so that readers can comprehend it better.
> > >
> > > Although the current scale of quantum hardware can not support the application of large-scale quantum algorithms, it still can not stop the widespread attention of academia and industry to quantum computing, especially in quantum machine learning [a,b]. We hope that our model can provide technical guidance for future research, and better exploit the immense benefits of quantum computing even in a classic simulation condition.
> > >
> > > Apart from quantum speedup, there are additional contributions of our proposed model, which have been illustrated in Line 66 to Line 80 in the introduction section. And we hope that this novel graph learning model can bridge the gap between graph neural tangent kernel methods and attention mechanism. Here we briefly give an illustration.
> > >
> > > 1. Better performance compared with state-of-art graph models. The numerical results in Table 1 and Table 2 demonstrate that our model
> > > (AttentionGNTK) outperforms GNTK on 7 out of 10 datasets, reflecting that the introduced infinite-width multi-head attention is useful and can better capture distinct properties between different graphs. Furthermore, we compare our model with another quantum-inspired graph learner QS-CNN. Our model surpasses QS-CNN on 4 out of 6 datasets and performs similarly on the rest 2 datasets. It shows the superiority of our model in quantum graph learning.
> > >
> > > 2. More robust compared with GNNs when the number of layers becomes larger. As illustrated in Table 5, when the number of layers is larger than 5 (but less than 10), Our model is more resistant to the oversmooth problem than GIN.
> > >
> > > 3. Fewer layers for reaching the peak of classification accuracy compared with GNTK. As illustrated in Table 6, our model (attentionGNTK) reaches the peak of classification accuracy when the number of layers is small, while GNTK needs more layers to reach, indicating that our model is easier to capture the global structure information of the graph.
> > >
> > > [a] Mernyei P, Meichanetzidis K, Ceylan I I. Equivariant quantum graph circuits, ICML, 2022.
> > >
> > > [b] Georgescu I. Quantum enhancement in generative models. Nature Reviews Physics, 2022.

---

> > > ### Author Response · Authors · 2022-08-09
> > > **Response to Reviewer DbrA Part 1**
> > >
> > > Thanks for your reply. We hope that our clarifications could address your questions. Realizing that your remaining confusion may be due to the limited space for our paper which is an intersection among emerging machine learning techniques (e.g. the infinidte width GNN) and quantum computing, we add more details in the appendix part to make our paper more self-contained and informative for you to make further evaluation. While we are thankful that the novelty and the potential impact of the paper are well acknowlede by the reviewers.
> > >
> > > **Q1: In the end I am still confused about what exactly the infinite width is, and it is not explained in the main paper.**
> > >
> > > **A1:** Thanks for your question. Graph neural tangent kernel (GNTK) is a generalization of NTK from infinite-width fully connected neural network to graph neural network, which is a well-established approach in recent literature and thus we miss some detailed explanation in our paper for space saving (and also due to the complexity of the details which otherwise will cost a lot of space), and the details can be found in [a]. Here we give an intuitive illustration and hope that this would be helpful for your understanding. We now also add these content in the appendix for your information which also makes the paper more self-contained.
> > >
> > > Consider a general GNN with the neighborhood feature aggregation function
> > >
> > > $$\mathbf{\hat{h}}\_{u}^{l} :=  \sum_{v \in \mathcal{N}(u) \cup\{u\}} \mathbf{h}_{v}^{(l-1)},$$
> > >
> > > and the central node feature update function ($R$ fully connected layers)
> > >
> > > $$\mathbf{h}\_{u}^{l} := \sqrt{\frac{c_{\sigma}}{d^{l}\_{R}}}\sigma\left(\mathbf{W}\_{R}^{l}\sqrt{\frac{c_{\sigma}}{d^{l}\_{R-1}}} \sigma\left(\mathbf{W}\_{R-1}^{l} \cdots \sqrt{\frac{c_{\sigma}}{d^{l}_{1}}} \cdot \sigma\left(\mathbf{W}\_{1}^{l}\mathbf{\hat{h}}\_{u}^{l}\right)\right)\right),$$
> > >
> > > where $\mathbf{h}\_{u}^{l}$ denotes the feature vector of node $u$ in the $l$-th layer, $\mathcal{N}(u)$ denotes the neighborhood of node $u$, $\mathbf{W}\_{r}^{l} \in \mathcal{R}^{d^{l}\_{r}\times d^{l}\_{r+1}}$ is the weight matrix ($r=1,\ldots,R$), $c_{\sigma}$ is a scaling factor. **The infinite-width of GNN means that the output dimension $d^{l}_{r}$ of  $\mathbf{W}\_{r}^{l}$ goes to infinity for $r=1,\ldots,R$ and $l=1,\ldots,L$.**
> > >
> > > Apart from the infinite-width GNN, the infinite-width transformer, which is exploited by us to enhance the GNTK, has also been investigated in [b]. Here we briefly explain what an infinite-width transformer looks like. Consider a single attention layer
> > >
> > > $$\mathbf{Q}^{h}=\mathbf{H}\mathbf{W}\_{Q}^{h}, \mathbf{K}^{h}=\mathbf{H}\mathbf{W}\_{K}^{h},\mathbf{V}^{h}=\mathbf{H}\mathbf{W}\_{V}^{h},$$
> > >
> > > $$\mathbf{G}^{h}=\frac{\mathbf{Q}^{h} {\mathbf{K}^{h}}^{\top}}{\sqrt{s}}, \widehat{\mathbf{H}}^{h}=\zeta\left(\mathbf{G}^{h}\right) \mathbf{V}^{h},$$
> > >
> > > in Eq.10, where $\mathbf{H} \in \mathcal{R}^{n\times d}$ is the node feature matrix and we ignore the superscript and the subscript for simplicity. $\mathbf{Q}^{h}$, $\mathbf{K}^{h}$, $\mathbf{V}^{h} \in \mathcal{R}^{d\times d'}$ are weight matrices corresponding to Query, Key and Value, respectively. $s$ is a scaling factor. The square matrix $\mathbf{G}^{h} \in \mathcal{R}^{n \times n}$ can be viewed as a matrix whose element corresponds to the similarity of each pair of nodes. Then the operation $\widehat{\mathbf{H}}^{h}=\zeta\left(\mathbf{G}^{h}\right) \mathbf{V}^{h}$ transforms the node features of the last layer to the next layer depending on the node similarity. For a multi-head attention (transformer) layer, the equation is given as
> > >
> > > $$transformer(\mathbf{H})=concat(\widehat{\mathbf{H}}^{1}||\widehat{\mathbf{H}}^{2}|| \cdots ||\widehat{\mathbf{H}}^{H})\mathbf{W}_{O},$$
> > >
> > > where $\mathbf{W}_{O} \in \mathcal{R}^{Hd'\times d''}$ is the weight matrix. **The infinite-width transformer means that the output dimension $d'$ of $\mathbf{Q}^{h}$, $\mathbf{K}^{h}$, $\mathbf{V}^{h}$, the output dimension $d''$ of $\mathbf{W}_{O}$, and the number of heads $H$ go to infinity.**
> > >
> > > Notice that in our code it is not necessary to evaluate the infinite-width model by setting the output dimension (and the number of heads) as a large number, which has already been covered in our rebuttal of A2.
> > >
> > > We have revised the paper according to the comments for a more clear clarification, with revisions marked in blue.
> > >
> > >
> > > [a] Du S S, Hou K, Salakhutdinov R R, et al. Graph neural tangent kernel: Fusing graph neural networks with graph kernels. NeurIPS, 2019.
> > >
> > > [b] Hron J, Bahri Y, Sohl-Dickstein J, et al. Infinite attention: NNGP and NTK for deep attention networks. ICML, 2020.

---

> > > ### Author Response · Authors · 2022-08-10
> > > **We would love to hear your feedback on our further rebuttal**
> > >
> > > Dear Reviewer DbrA,
> > >
> > > As the discussion period is close to the end and we provide further clarifications for your remaining confusion, we wanted to reach out to see if our rebuttal response has addressed your concerns.
> > >
> > > We are more than happy to discuss further if you have any further concerns and issues, please kindly let us know your feedback. Thank you for your time and help!

---

> ### Author Response · Authors · 2022-08-02
> **Response to Reviewer DbrA Part 2**
>
> **Q3: From lines 75-76 it can be thought that the authors propose to use GAT, and thus the claim that it is non-local as opposed to the standard propagation of GCN is not in order. I suggest that the authors clarify that their suggestion is to use a transformer.**
>
> **A3:** Thanks for your suggestion that we agree and we will clarify it.
>
> **Q4: The authors claim a computational cost reduction from $O(N^2)$ to $O(N)$. However, when deploying a transformer that considers all pairs of nodes, it is not clear to me how the costs are saved.**
>
> **A4:** Thanks for your comments and we believe there may be some misunderstanding that we try to clarify for your rethinking. In a nutshell, our cost reduction result is not surprising seeing the recently well-established kernelized tranformer techniques in [a]. We give more details in below and will also clarify it in the paper.
>
> In the general case, transformer measures the similarity of each pair of nodes which could introduce huge computational overheads. It will get worse when the number of heads and the output dimension of Query, Key, Value become very large. We circumvent this obstacle by two steps which have been presented in the main paper.
>
> 1. As discussed in A2, NTK allows to implement and analyze some deep learning modules in terms of kernel methods, which is also applicable to the transformer. According to [a], the dynamics of the infinite-limit transformer can be captured by the NTK given in Eq.12. In order to adapt it to quantum parallelism, we modify it slightly and the modified version is Eq.13, under the assumption that in the limit of infinite width neural network the output converges in distribution to a multivariate normal with a block diagonal covariance [b]. The modified version indicates that computing an element of NTK $\mathbf\Theta(\mathbf{x},\mathbf{x}')$ depends only on this element in the same position of NTK in the previous layer. This will be a potential quantum speedup if the initial NTK can be encoded into the amplitudes of a quantum superposition state.
>
>
> 2. In Section 2.3, we propose the feature encoding and the estimation method for the initialized NTK. Thus, each element of initial NTK is encoded into a quantum superposition state given as
> $$\frac{1}{\sqrt{n n^{\prime}}}\sum_{u\in V} \sum_{u^{\prime}\in V^{\prime}}|u\rangle|u^{\prime}\rangle\left(\sqrt{P_{u u^{\prime}}}\left|0, g_{u u^{\prime}}\right\rangle+\sqrt{1-P_{u u^{\prime}}}\left|1, g_{u u^{\prime}}^{\prime}\right\rangle\right),$$
> where amplitude $P_{u u^{\prime}}$ is the estimation of NTK of the last layer, $|u\rangle|u^{\prime}\rangle$ is the basis of the quantum superposition state and $|1, g_{u u^{\prime}}\rangle$ denotes a garbage state. By acting the transformation given by Eq.17 on this state, all elements of NTK can be transformed simultaneously according to Eq.13. However, the prerequisite for this parallelization is to calculate the matrix $\widetilde{\mathbf{T}}_{\zeta}$ in advance. In the quantum domain, this is done by estimating the NTK between two same graphs instead of arbitrary pair of graphs, which requires $O(N)$ time.
>
> [a] Hron J, Bahri Y, Sohl-Dickstein J, et al. Infinite attention: NNGP and NTK for deep attention networks. ICML, 2020.
>
> [b] Novak R, Xiao L, Lee J, et al. Bayesian deep convolutional networks with many channels are gaussian processes. ICLR, 2019.
>
> **Q5: Figure 1 is not self contained and also not explained in text properly. It is not clear what 'SVC' and 'SVR' means.**
>
> **A5:** Thanks for your suggestion. SVC and SVR is the abbreviation of Support Vector Classification and Support Vector Regression, respectively. We hope to give readers an overview of quantum graph learning via Figure 1. We will make it more clear.

---

> ### Author Response · Authors · 2022-08-02
> **Response to Reviewer DbrA Part 1**
>
> Thanks for your insightful comments. We hope our answers below address all your concerns.
>
> **Q1: From an originality point of view, the concept of QNTK was already proposed in different papers that were not considered in the paper (see for example "Representation Learning via Quantum Neural Tangent Kernels" (arXiv:2111.04225) and "Quantum tangent kernel (arXiv:2111.02951)).**
>
> **A1:** Thanks for your attention to our work but we find these works are just literally similar to our work, but have nothing to do with ours. Specifically, these two papers do mention the term QNTK, but their definition is quite different from ours. The motivation of these two papers is to analyze the trainability and expressive power of variational quantum circuits through NTK. Briefly speaking, the authors' intention is to encode the classical data into quantum states and process them in the quantum Hilbert space by developing a specific quantum variational circuit. The tool they use to study the dynamics of quantum variational circuits is NTK, which is a powerful tool in the study of deep learning theory.
>
> In our work, QNTK is a metric measuring the similarity of two input graphs. In this context, NTK is the kernel that captures the dynamics of infinite-width GNNs, as well as the multi-head attention where the number of heads and the dimension of output go to infinity (see Eq.8,9,12,13). Our QNTK corresponds to a classical deep learning model, which naturally captures the structure information of the input (graph) data, instead of only analyzing the existing quantum deep learning model, i.e., quantum variational learning through NTK.
>
> We will add these two references and clarify such obvious differences in our revision. Thank you.
>
> **Q2: It is not clear what do the authors mean by an infinite width GNN. In the end, the method needs to be deployed in code, so does it mean that the number of channels is simply very large? It is not provided in text what the actual width of the network is.**
>
> **A2:** Thanks for your comments and we will give a more clear illustration of the definition and the implementation of infinite-width GNN.
>
> In fact, infinite-width GNN is a well established term in the GNN literature [a,b] (hence we miss some detailed explaination in our paper). Specifically, consider a GNN given in Eq.1 and Eq.2, the infinite-width GNN means the output dimension of learnable weights matrices $\{\mathbf{W}_{r}^{l}\}$, where $r=1,\ldots,R, l=1,\ldots,L$, go to infinity. Moreover, the infinite-width transformer means the output dimension of the transformer layer and the number of heads go to infinity, which is already covered in line 174.
>
> In practice, the actual implementation of infinite-width GNN and the infinite-width transformer does not need to give the number of neurons, or the number of channels in each layer. One of the theoretical cornerstones of our work is NTK [c]. NTK is a powerful tool for studying the dynamics of DNN, avoiding us from defining a very wide output in the experiment. Based on this, GNTK is designed to analyze and train infinite-width GNN. When in the infinite situation, neighborhood feature aggregation (Eq.1), central node feature update (Eq.2) and transformer (Eq.11) can be recovered by our proposed kernel AttentionGNTK. Thus, we can implement the AttentionGNTK in our code to analyze the infinite-width GNN. The code to generate the NTK of infinite-width GNN is as follows:
> ```
> sigma, dot_sigma, diag = self.__next_diag(sigma)
> diag_list.append(diag)
> ntk = ntk * dot_sigma + sigma
> nngp, diag_nngp = self.__next_diag_nngp(nngp)
> nngp_old = np.copy(nngp)
> nngp = np.matmul(np.matmul(nngp, nngp), nngp.T)
> ntk = 2 * nngp + np.matmul(np.matmul(nngp_old, ntk*2), nngp_old.T)
> ```
>
> [a] Du S S, Hou K, Salakhutdinov R R, et al. Graph neural tangent kernel: Fusing graph neural networks with graph kernels. NeurIPS, 2019.
>
> [b] Huang W, Li Y, Du W, et al. Towards Deepening Graph Neural Networks: A GNTK-based Optimization Perspective. ICLR, 2022.
>
> [c] Jacot A, Gabriel F, Hongler C. Neural tangent kernel: Convergence and generalization in neural networks. NeurIPS, 2018.

---

> ### Author Response · Authors · 2022-08-07
> **We would love to hear your feedback on our rebuttal**
>
> Dear Reviewer DbrA,
>
> Thanks again for your review. We hope our answers could increase your confidence. As the discussion period is close to the end and we have not yet heard back from you, we would be glad to see if our rebuttal response has addressed your concerns questions/concerns.
>
> We are more than happy to discuss further if you have any further concerns and issues, please kindly let us know your feedback. Thank you for your time and help!

---

> ### Author Response · Authors · 2022-08-09
> **Looking forward to your reply**
>
> Dear Reviewer DbrA,
>
> Thanks again for your review.  Since the discussion period is approaching its end, we would be glad to hear from you if our response has properly addressed your questions/concerns. If you have further questions, I will be happy to provide clarifications.
>
> kind regards

---

### Author Response · Authors · 2022-08-02
**General Response by Authors**

Dear area chair and reviewers,

We thank all the three reviewers for your valuable reviews that have helped improve and revise our submission. We are happy that the theoretical novelty of studying the learning efficiency problem of deep GNN through Quantum Graph Neural Tangent Kernel has been recognized by all three reviewers. The main concerns include missing references on related works, unclear presentations and inaccurate notations. Besides, additional experiments and numerical results analysis are required. For all the questions that need to be clarified, we provide detailed response one by one. We have revised the paper according to the comments, with revisions marked in blue.

---

### Meta-Review · Area_Chair_jMYU · 2022-08-26

**Recommendation:** Accept
**Confidence:** Certain

**Metareview:**

The authors propose a Quantum Neural Tangent Kernels (QNTK) approach for GNNs. The paper shows promising results of using quantum computing to speed up computations for graph data.  This is a novel idea, and the paper presents solid theoretical and empirical evaluations. Thus the AC recommends acceptance.

**Award:**

No

---

### Decision · Program_Chairs · 2022-09-14

Accept